# Changes in the Spore Proteome of *Bacillus cereus* in Response to Introduction of Plasmids

**DOI:** 10.3390/microorganisms10091695

**Published:** 2022-08-24

**Authors:** Xiaowei Gao, Bhagyashree N. Swarge, Winfried Roseboom, Yan Wang, Henk L. Dekker, Peter Setlow, Stanley Brul, Gertjan Kramer

**Affiliations:** 1Molecular Biology & Microbial Food Safety, Swammerdam Institute for Life Science, University of Amsterdam, 1098 XH Amsterdam, The Netherlands; 2Mass Spectrometry of Biomolecules, Swammerdam Institute for Life Science, University of Amsterdam, 1098 XH Amsterdam, The Netherlands; 3Molecular Biology and Biophysics Department, UConn Health, Farmington, CT 06030-3305, USA

**Keywords:** germinant receptor, fluorescent proteins, proteomics, effects of plasmids

## Abstract

Fluorescent fusion proteins were expressed in *Bacillus cereus* to visualize the germinosome by introducing a plasmid that carries fluorescent fusion proteins of germinant receptor GerR subunits or germinosome scaffold protein GerD. The effects of plasmid insertion and recombinant protein expression on the spore proteome were investigated. Proteomic analysis showed that overexpression of the target proteins had negligible effects on the spore proteome. However, plasmid-bearing spores displayed dramatic abundance changes in spore proteins involved in signaling and metabolism. Our findings indicate that the introduction of a plasmid alone alters the spore protein composition dramatically, with 993 proteins significantly down-regulated and 415 proteins significantly up-regulated among 3323 identified proteins. This shows that empty vector controls are more appropriate to compare proteome changes due to plasmid-encoded genes than is the wild-type strain, when using plasmid-based genetic tools. Therefore, researchers should keep in mind that molecular cloning techniques can alter more than their intended targets in a biological system, and interpret results with this in mind.

## 1. Introduction

*B. cereus* is a Gram-positive, facultative anaerobic, spore-forming human pathogen, well known as a causative agent for food spoilage and food-borne illnesses, both diarrheal and emetic [1]. The removal of this organism from the food chain is difficult due to its ability to form spores. When the environment is not favorable, *B. cereus* vegetative cells divide asymmetrically and ultimately release spores in a process called sporulation [2]. Spores are dormant and highly resistant to extreme conditions, such as high temperature, radiation, desiccation and so on, making them difficult to kill. These resistance properties are mostly due to the composition of the spore core and, to a lesser extent, the multi-layered structures that encase the core [3]. The spore core, the innermost layer, contains DNA, RNA and most enzymes, and DNA is protected by its saturation with small, acid-soluble spore proteins (SASPs) against heat and toxic chemicals [4]. The spore core also contains a high level of a 1:1 chelate of dipicolinic acid (DPA) with Ca^2+^ (CaDPA), which also contributes to spore resistance [3]. The spore core is surrounded by the inner membrane (IM), which is highly compressed and has low permeability to small molecules [5,6]. Outside the IM is a germ cell wall layer, the peptidoglycan composition of which is identical to that of the growing cell wall. The spore cortex peptidoglycan layer, outside the germ wall, has several cortex-specific modifications that allow for enzyme recognition and cleavage when spores germinate. The cortex layer is required for spore core dehydration and dormancy [7]. Above the cortex is the outer membrane and then the proteinaceous coat layer, which protects the spore against chemicals and enzymes. In *B. cereus* spores, the outermost layer is the exosporium, which may have a role in pathogenesis [2].

Although the spore is dormant, it monitors the environment, and low-molecular-weight nutrients can be sensed by germinant receptors (GRs) located in the IM. Once committed to germination, spores release CaDPA from the spore core, degrade the spore cortex and then the coat layer and grow out into vegetative cells, losing their extreme resistance properties along the way [3]. The growing cell then produces toxins, resulting in food spoilage or poisoning. Hence, potential strategies for spore eradication are to trigger germination to easily kill germinated spores or to prevent germination, all to avoid toxin production [1]. To improve and develop new strategies for spore eradication, more detailed knowledge about spore germination processes is required, especially about the GRs that are expressed at only low levels in spores [8]. The *B. cereus* GR GerR complex consists of three subunits, A, B and C, encoded in a single operon [9]. Subunits A and B have multiple transmembrane domains, making their extraction, solubilization and purification challenging, although the GerR C subunit is much more hydrophilic, but with a lipid anchor. The GerD lipoprotein is also critical for rapid GR-dependent germination [10], and is thought to interact with GRs in an IM complex termed the germinosome [11,12].

The green fluorescent protein (GFP) was first discovered and isolated in the early 1960s [13]; such fluorescent proteins are now widely used as fusion tags to monitor protein localization in living organisms and the overexpression and purification of membrane proteins. Taking advantage of this technology, localization of GerR and the visualization of germinosomes in *B. cereus* were reported after genetically fusing the fluorescent proteins to the C-termini of GR proteins to act as reporters, followed by microscopic detection in intact spores [11,12].

Although fluorescent proteins are a vital tool in molecular studies, their introduction into a biological system can have a variety of unexpected and undesired effects. Apart from their importance in validating whether a fusion protein localizes and functions similarly to its endogenous counterpart [14,15], the introduction of foreign genes through genetic tools can potentially cause a variety of unintended effects in both pro- and eukaryotes [16,17,18]. Well-known side effects of recombinant protein production shown by studies in *Escherichia coli* are a significant plasmid metabolic burden, altering the expression of genes of central metabolism, as well as marked increases in expression levels of heat shock proteins [19,20]. To visualize GRs and elucidate how they associate in *B. cereus* spores, a variety of plasmids containing one or combinations of GR genes fused to fluorescent proteins were introduced into wild-type *B. cereus* [11,12]. We were thus interested in whether the introduction of plasmids bearing fusion proteins would affect the global protein content of the mature spore in any way. To study this, we used label-free quantitative proteomics and mass spectrometry (MS) to profile the protein content of mature spores from *B. cereus* transformed with different plasmids containing fusions of GRs to fluorescent proteins, an empty plasmid, a plasmid with GFP and a chromosomal deletion of the *cotE* gene encoding a major spore coat protein [21]. We quantitatively compared the protein content of spores with these plasmids to that of the plasmid-free spores of the wild-type (WT) strain (Table 1). In this manner, we were able to make a distinction between the effects of plasmid introduction, heterologous protein production such as of GFP and the synthesis of specific GR fusion proteins on the protein composition of mature spores.

## 2. Materials and Methods

### 2.1. Strain and Culture Conditions

The WT strain and its derivatives used in this study was *B. cereus* ATCC 14579, and construction of the derivatives has been described previously [11,12,21]. The details of bacterial culture and spore harvest were also described previously [12,22]. In brief, a single colony was inoculated and grown aerobically at 30 °C overnight in tryptic soy broth (TSB) with 275 µg/mL spectinomycin or 10 µg/mL erythromycin (Sigma-Aldrich Chemie B. V., Zwijndrecht, The Netherlands). Then, cells were spun down and cultured in a chemically defined growth and sporulation (CDGS) medium without antibiotics for 96 h, which gave a culture that contained ~99% dormant spores. The spores were harvested by centrifugation and washed with 0.1% Tween-80 in cold Milli-Q water at least four times to remove any small number of growing cells or sporulating cells that had not lysed, as well as macromolecules released from lysed sporulating cells.

### 2.2. Sample Preparation and LC-MS/MS Analysis

Spore samples in 1 mL water at an OD600 = 2 were processed using the “One-Pot” method [23]. To improve identification depth, each WT sample was also separated into 10 fractions using zwitterionic (ZIC)–hydrophilic interaction liquid chromatography (HILIC)-based peptide fractionation [23]. Around 200 ng peptides for each sample were dissolved in 0.1% formic acid for mass spectrometry analysis. All samples had three biological replicates. Mass spectrometry analysis of all samples was performed as described previously [24].

### 2.3. Data Analysis

Raw MS/MS data were searched in Maxquant (version: 1.6.14.0) [25] against the *B. cereus* ATCC 14579 database downloaded from Uniprot; to estimate the false spectrum assignment rate, a reverse version of the same database was also searched. Settings were default for timsDDA. Trypsin/P was selected as the digestion enzyme, with a maximum of 2 missed cleavages. The Oxidation (M) was set as a variable modification and Carbamidomethyl (C) as a fixed modification. The “Match between runs” was selected with a matching time window of 0.2 min and a matching ion mobility window of 0.05 indices. For label-free quantification, both intensity-Based Absolute Quantitation (iBAQ) and Label-Free Quantitation (LFQ) were enabled. The “LFQ min. ratio count” was set as 2, and it was set as 1 for the quantification of GerRB.

Data analysis was performed in Perseus (version: 1.6.15.0) [26]. If not mentioned specifically otherwise, LFQ intensities were used for data analysis. Data were filtered based on valid values with “min 2” “in at least one group”. The second replicate of Sample M009 was removed, since it was found to be an outlier. The functions of proteins were categorized according to Kyoto Encyclopedia of Genes and Genomes (KEGG) pathway [27] analysis. The LFQ intensities were used for a volcano plot using a limma R package for determining significant changes in spore protein levels. Spores of strains *cotE* and M001 were compared with WT spores, and other mutant spores were compared with M001. Data were obtained with z-scoring of rows, which corresponded to protein expression levels over the different samples in the series. LFQ intensities were normalized by subtracting the median protein expression level per sample and z-scoring of rows as mentioned above, before examining the abundances of GR subunits, GerD and CotE.

## 3. Results

### 3.1. The Influence of Molecular Cloning on the Spore Proteome

The strains used in this study are listed in Table 1. The deletion of cotE was constructed by allele exchange and verified by DNA sequencing [21]. The cotE null strain was used as a positive control for large-scale changes to the spore proteome composition versus that of WT spores. Strain M001 contains only an empty plasmid to check for transformation and plasmid maintenance effects. Strains M003 and M006 contain a plasmid with the gerR promotor from the B. cereus genome controlling the expression of either only the GerRB subunit fused to the super folder derivative of GFP (sGFP) or the entire gerR operon (gerA-gerB-gerC) of which gerRB, the last gene in the gerR operon, is fused to sGFP. Strain M005 carries a plasmid with the constitutive aminoglycoside phosphotransferase (aph) A3′ promotor and sGFP to check the effect of constitutive sGFP expression. Strain M007 contains gerD fused to the gene for the red fluorescent protein mScarlet controlled by the gerD promotor. Strains M008 and M009 have either gerRB-sGFP or the complete gerR operon (with gerRB-sGFP) under gerR promotor control plus gerD-mScarlet controlled by the gerD promotor.

**Table 1 microorganisms-10-01695-t001:** Strains used in this study.

Strains	Relevant Genotype	Source
*B. cereus* WT	*B. cereus* ATCC 14,579 wild type	lab stock
*B. cereus cotE*	*B. cereus ∆cotE* deletion mutant Sp*^r^*	[21]
*B. cereus* M001	*B. cereus* carrying pHT315 Ery*^r^*	[12]
*B. cereus* M003	*B. cereus* carrying pHT315-P*gerR*-*gerRB*-*sGFP2* Ery*^r^*	
*B. cereus* M005	*B. cereus* carrying pHT315-P*aphA3′*-*sGFP2* Ery*^r^*	
*B. cereus* M006	*B. cereus* carrying pHT315-P*gerR*-*gerR*-*sGFP2* Ery*^r^*	
*B. cereus* M007	*B. cereus* carrying pHT315-P*gerD*-*gerD*-*mScarlet* Ery*^r^*	
*B. cereus* F09 (M008)	*B. cereus* carrying pHT315-P*gerR*-*gerRB*-*sGFP2*I	[11]
	-P*gerD*-*gerD*-*mScarlet* Ery*^r^*	
*B. cereus* F06 (M009)	*B. cereus* carrying pHT315-P*gerR*-*gerR*-*sGFP2*	
	-P*gerD*-*gerD*-*mScarlet* Ery*^r^*	

Sp*^r^*, spectinomycin resistance; Ery*^r^*, erythromycin resistance.

The spore proteome of these different strains was investigated to estimate the effects caused by recombinant gene expression. In total, 3545 proteins were identified in the whole set of samples, among which 3323 proteins were identified in at least two replicates (Appendix A). As shown in Figure 1A, over 2600 spore proteins were quantified from the biological replicates of the different strains. The average coefficient of variation (CV) between replicates was below 0.05, showing high quantitative reproducibility between replicates (Figure 1B).

To detect significantly altered content of proteins in spores, a limma test was conducted on pairwise groups, and the difference in expression is shown in a volcano plot (Figure 2). Proteins with a log_2_ (fold change) ≥ |±1| and −log(adjusted *p*-value) ≤ 0.01 were considered to be significantly differentially present in the spore. First, we compared values in both the *cotE* deletion strain and M001 (empty plasmid) to the WT spores. Compared to WT, the deletion of *cotE* resulted in 801 proteins significantly down-regulated and 543 proteins significantly up-regulated. Introduction of an empty plasmid caused 993 spore proteins to be significantly down-regulated and 415 proteins were significantly up-regulated (Figure 2, Appendix A). The differentially expressed proteins in the *cotE* strain and M001 compared to WT were quite evenly distributed over global KEGG pathways (Figure 3); only the KEGG term “Unclassified: signaling and cellular process” containing spore-specific proteins had slightly more protein abundance changes in M001 spores than in *cotE* spores. The large number of alterations in spore protein levels by the introduction of an antibiotic resistance marker-bearing plasmid was surprising, and had an effect on the protein content of the mature spore on a par with that of a deletion of a major spore coat protein.

The introduction of plasmid alone induced many alterations in spore protein content. To identify changes induced by specific GR fusion protein expression (strains M003, M006 and M007), combinations of two GR fusion proteins (strains M008 and M009) or the constitutive expression of a fluorescent protein (M005), we compared these spores’ proteomes to that of empty plasmid-containing spores (M001). As is obvious, this resulted in far fewer altered proteins (see below) and underscored the necessity of using a proper control (not a wild-type plasmid-free strain) when performing global studies on strains altered through introduction of a plasmid. When we looked at additional effects from the constitutive expression of sGFP, there were only three proteins less abundant and one protein more abundant compared to empty plasmid-containing spores. One of the latter proteins (Hemolysin BL binding component) was also down-regulated and one (tRNA(Ile)-lysidine synthase) was up-regulated in four out of five of the other strains bearing plasmids containing fusion proteins. No other proteins were found in more than one or two of the other strains as being regulated in the same manner. Together with the relatively small number of additional altered proteins, this suggested that expressing a recombinant GR-fusion protein had relatively minor additional effects on the spore protein composition compared to the introduction of the plasmid alone.

Similarly, the numbers of alterations in protein content in spores containing a single GR fusion protein—M003 (1 down), M006 (8 up, 9 down), M007(19 up, 8 down)—were markedly smaller than plasmid introduction and only slightly higher in three strains compared to the constitutive expression of sGFP. Among the proteins of altered abundance in spores expressing the entire *gerR* operon (GerRB-sGFP, M006) and GerD-mScarlet-1 (M007), there were three proteins related to sporulation (sporulation-specific protease YabG in M007) and germination pathways (GerA and GerC family proteins in M006). In the two strains which had plasmids expressing both GerRB-sGFP and GerD-mScarlet-1 (M008) or the entire *gerR* operon giving GerRB-sGFP as well as GerD-mScarlet-1 (M009), there were a larger number of proteins that showed an altered abundance in M008 (26 up, 42 down) and M009 (16 up, 53 down) compared to the empty plasmid spores. Of these, 17 proteins correlated in abundance changes between the two strains compared to an empty plasmid (Appendix A), but relatively few of these proteins seemed directly related to sporulation or germination, suggesting that no large changes in the spore protein content related to these pathways were induced by the fusion proteins’ expression. In future work, it would be worthwhile to compare the spore germination and sporulation of strains with and without an empty plasmid in detail; indeed, preliminary results suggested that the presence of an empty plasmid slowed sporulation slightly (data not shown).

### 3.2. Examining Relative Levels of Spore Proteins in the Different Strains

Strains were constructed in this work to study signaling events relating to spore germination, as it is of particular interest to observe how genetic alterations influence the expression of spore proteins. As stated above, the recombinant strains produce a fluorescent fusion protein of either only GerRB (M003), the entire GerR GR and with GerRB fused to sGFP (M006) or with GerD fused to mScarlet (M007). The other two strains are a combination of one of the first two with GerD fused to mScarlet on the same plasmid (M008 and M009). The relative abundances (z-scored normalized LFQ values) of the GerR protens, GerD and CotE are shown in Figure 4. In the *cotE* mutant strain, three *cotE* peptides were still detected due to the match-between-run algorithm of Maxquant, but these were likely due to carry-over between samples and not the presence of CotE in the mutant strain. When the WT spore levels of the GerD and three GerR proteins were compared to those in spores of strains carrying an empty plasmid (M001) or the plasmid constitutively expressing sGFP (M005), there was a clear reduction in the recombinant strains, which did not seem to change much due to sGFP expression. Compared to M001 and M005 spores, the expression of CotE in the other strains was relatively constant, while in the strains where only GerRB was expressed from the plasmid (M003 and M008 in combination with GerD), there was a small increase in its level, while the other proteins of the *gerR* operon remained unaffected when compared to strain M001. However, when the entire *gerR* operon was expressed, there was a marked increase in the levels of all three GerR subunits (M006 and M009 vs. M001). GerD expression from a plasmid (M007 and M009) increased its own expression level compared to strain M001, while its expression seemed not to influence the expression of GerR proteins (M007 and M008 vs. M001).

The effect of expressing the *gerR* operon or *gerD* from a plasmid on the abundance of proteins in spores involved in germination and sporulation is shown in Figure 5A. Based on the results from Figure 2, significantly altered proteins are marked with an asterisk. Among these, GerK, GerS, GerQ, GerI, GerB and GerL are GRs that respond to different nutrients [9], and SpoVA proteins comprise a channel involved in CaDPA release in germination. The coat protein GerP is involved in transport of signals to GRs [28]. Gpr is the enzyme that degrades SASPs following germination. The cortex lytic enzymes SleB and CwlJ are involved in the degradation of the cortex, and coat proteins YpeB and GerQ are required for SleB/CwlJ stability [29]. Again, compared with WT spores, most of these proteins were markedly less abundant in empty plasmid-containing spores (M001), as much or more so than in a *cotE* deletion mutant. On the other hand, spore abundances of GerK, CwlJ and proteins in the coat layer (GerP, GerQ) were not significantly affected. The transcription of GR operons is regulated by sigma factor SigG [30] and repressor SpoVT [31]. Compared with WT spores, the abundances of these two transcription regulators were also reduced in spores of strain M001. However, as is clear from Figure 2, the additional expression of different proteins from the plasmid did not seem to induce further alterations in protein abundance in mature spores, GerLC being the only significantly down-regulated protein found in spores of M008 compared to M001 spores.

We also looked at the influence of empty plasmid transformation and *cotE* deletion on the composition of the spore coat layer and SASPs in the spore core (Figure 5B,D). The effects of *cotE* deletion on spores were studied previously [22]. Consistent with the previous study [22], the abundance of CotE-independent proteins (CotH, CotG, CotS, CotD) was significantly increased, while abundances of CotE-dependent proteins, such as SafA, decreased. Most coat proteins were significantly down-regulated in M001 versus WT spores, except CotH, CotD, CotG, CotW, CotS2 and BC5056. Apart from SasP-1, the amounts of SASPs were significantly down-regulated in M001 spores. The SspE protein has been reported to be CotE-dependent [22], but was not identified in the current work. The levels of the SASP SspO were significantly decreased in the *cotE* deletion strain, suggesting its CotE dependence. Overall, these data again underscore that WT strains are not apt controls when using plasmid-based systems to study processes in spores.

### 3.3. Plasmid-Induced Physiological Changes Are Apparent in the Spore Proteome

Prior research has reported that the metabolic burden of a plasmid in bacteria can alter the expression of genes of central metabolism as well as heat shock proteins [19,20]. Dormant spores, having no metabolic activity themselves, do carry with them proteins to restart metabolism upon germination. Furthermore, the vegetative proteins found in spores can be thought of as a representation of the metabolic state that the developing spore was in when the spores were forming [32]. For example, it has been demonstrated that spores have different properties in response to different temperatures during sporulation [33,34,35]. As shown in Figure 3, many of the proteins altered in the spore proteome comparing spores of strain M001 carrying an empty plasmid to plasmid-free WT spores belonged to central metabolic pathways. The quantitative information on glycolysis, tricarboxylic acid (TCA) cycle and pentose phosphate (PP) pathway proteins found in spores is shown in Figure 6. Compared with WT spores, the introduction of an empty plasmid resulted in higher levels of TCA cycle proteins. The key enzyme citrate synthase (CS) in the cycle was significantly up-regulated, as were the enzyme levels in the pathway producing acetyl-CoA from acetate and ethanol. In contrast, there was a reduction in proteins of the PP pathway in spores. Glycolytic enzymes were mostly decreased, apart from GAPDH and PGAM, showing that alterations in central metabolism induced by the presence of plasmids are apparent in the spore proteome.

The abundance of heat shock proteins and heat-induced proteins (DnaJ, DnaK, ClpC, ClpX, GrpE) [36] in the proteome of M001 and WT spores is shown in Figure 5B. Compared to WT, only two small heat shock proteins (SHSP1, SHSP2) showed increased abundance, while ClpX, DnaJ, Lon and HslO were down-regulated in M001 spores. The small heat shock proteins were also found to be up-regulated in *cotE* spores, but ClpX and DnaJ levels were not significantly affected. Compared to M001 spores, the abundances of heat shock proteins in M003-M009 spores were not significantly affected (Appendix A), showing again that plasmid introduction has a far larger effect on the spore proteome compared to recombinant fusion protein expression.

## 4. Discussion

There are many studies on the influence of environmental conditions during sporulation on spore composition. In previous studies in *B. cereus*, damage of the exosporium was observed in spores formed at high temperatures [37], and differences in spore coat protein composition were observed when sporulation occurred at 20 °C or 37 °C [21]. The concentrations of minerals accumulated in the spore core and the composition of cortex peptidoglycan are also affected by the composition of the sporulation medium [5,38]. However, little is known about the global response of *B. cereus* to perturbations in the intracellular environment. Consequently, in this work, we investigated alterations in the spore proteome induced by plasmid introduction and the expression of plasmid-encoded proteins. These experiments were conducted using *B. cereus* mutant strains constructed to localize and visualize protein–protein interactions between the multi-subunit GR GerR and the scaffold protein GerD in *B. cereus* spores.

Analysis of the proteome of spores expressing proteins on different plasmids showed that overexpression of GR subunit GerRB alone only marginally increased the abundance of GerRB itself, and with no effect on levels of the other two GerR subunits. In contrast, expression of all three GerR subunits from a plasmid did result in marked increases in the levels of all three GerR subunits in the spore proteome. These results suggest that the three GerR subunits could be assembled together during sporulation, while excess free GerRB may be degraded. However, studies on the GerA GR in *B. subtilis*, a homologue of GerR, showed that when the A or C subunit alone was overexpressed [39,40], spores showed similar germination behaviors as with overexpression of the whole *gerA* operon [41]. Our results on *gerR* operon expression suggest that GerR assembly in *B. cereus* may differ from that of GerA in *B. subtilis*. By homology with *B. subtilis*, the *B. cereus gerR* operon seems likely to be a member of the SigG regulon [30] and putative SigG binding sites are found in upstream regions of *B. cereus* GR operons [42]. In *B. subtilis*, the SigG regulon includes other spore proteins, including GRs, GerD and SASP, all of which are synchronously expressed [43]. In our dataset, overexpression of the *gerR* (M006 and M009) operon did not affect levels of other likely members of the SigG regulon, such as GerD and other GRs, indicating that there was no feedback regulation of SigG regulon expression in our *B. cereus* strain [43].

The largest effect that we found on the proteome of mature spores was from the introduction of an empty plasmid containing only the antibiotic resistance marker used for selection. The number of proteins whose levels were significantly altered by such a plasmid was slightly larger than the number altered by a *cotE* deletion compared to wild-type spores. Proteins whose level was altered by introduction of a plasmid ranged from those of central carbon metabolism, small heat shock proteins and many spore-specific proteins including GRs. As spores are metabolically dormant, their protein levels likely reflect the state of the developing spore during sporulation. Introduction of plasmids having effects on central carbon metabolism and heat shock proteins has been reported before in organisms such as *E. coli* [16,19,20,44]. However, we now report their effects on the spore proteome for the first time.

When the effects of the introduction of fluorescent fusion proteins on a plasmid to that of an empty plasmid vector on spore proteomes were compared, additional effects of the fluorescent fusion proteins on the spore proteome were minor. Indeed, introduction of an empty plasmid caused differential abundance in 1293 proteins (916 of which were down-regulated), amongst which are many spore coat proteins (Appendix A). This latter finding may explain the observation that spore autofluorescence, which has been attributed to coat proteins in *B. subtilis* spores [45], was reduced in *B. cereus* M001 spores compared to WT spores [12]. A limitation of our current study is that we cannot differentiate between plasmid introduction and antibiotic resistance marker effects. When interpreting differences in the spore proteome of the *cotE* deletion strain compared to wild-type spores, a possible confounding factor is the antibiotic resistance marker of the deletion strain. Overall, our results indicate that an empty plasmid (carrying the same selection marker) is a more apt comparison than a wild-type sample when spore proteomes are to be compared. Furthermore, researchers should keep in mind that molecular cloning can influence more than only its target genes, and results from such work should be interpreted with care.

## Figures and Tables

**Figure 1 microorganisms-10-01695-f001:**
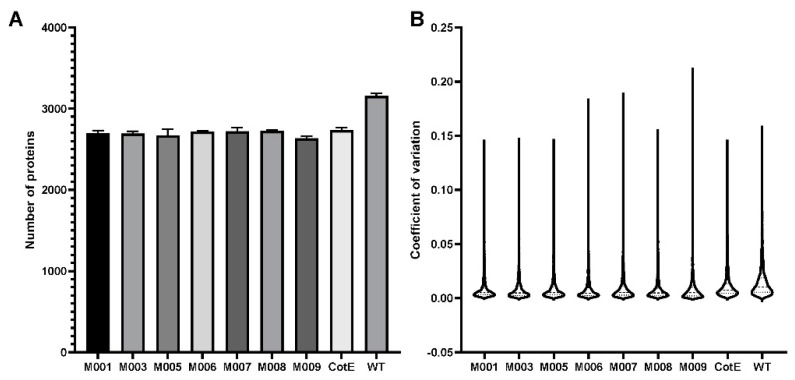
(**A**): The number of identified spore proteins in each strain. (**B**): Coefficient of variation between replicates of each strain.

**Figure 2 microorganisms-10-01695-f002:**
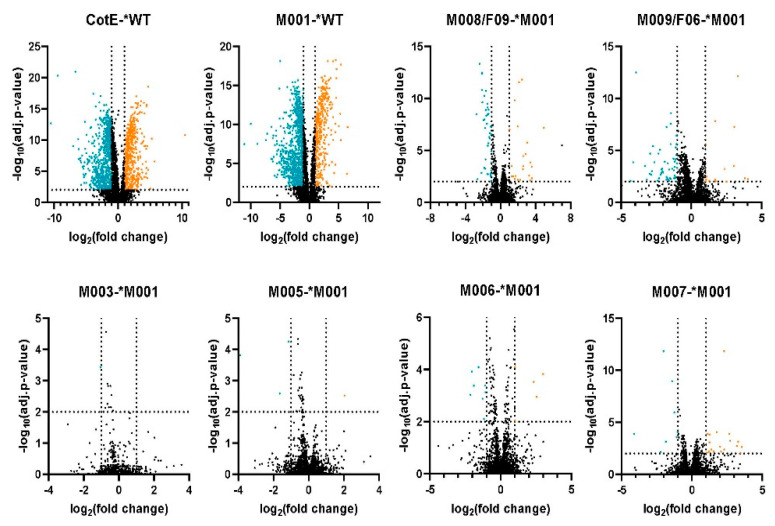
Volcano plots of t-test proteomes between spores of different strains. * denotes control group. The orange dots represent significantly up-regulated proteins, and blue dots indicate significantly down-regulated proteins compared with the control group. The vertical dotted lines indicate two-fold change. The horizontal dotted line represents *p* = 0.01.

**Figure 3 microorganisms-10-01695-f003:**
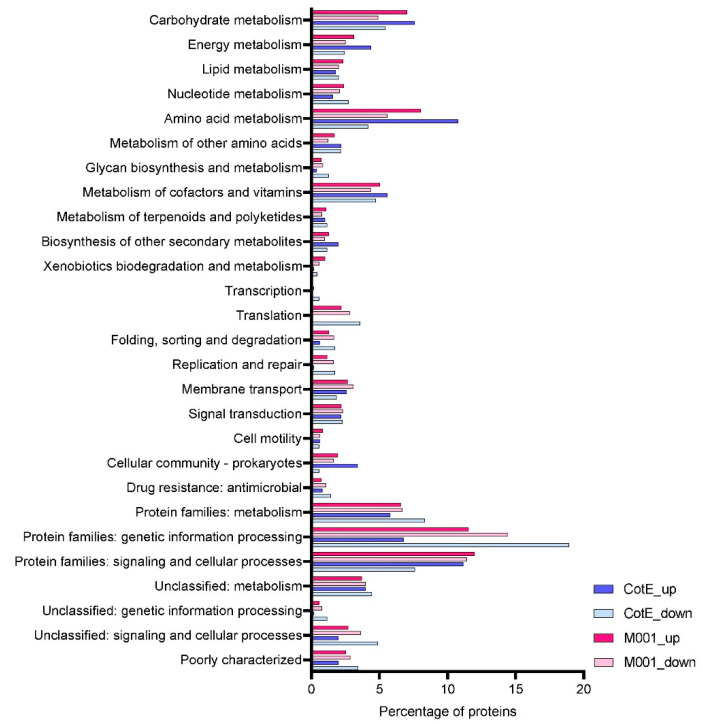
The KEGG functional enrichment of differentially expressed proteins in spores of strains *cotE* and M001 compared with WT spores.

**Figure 4 microorganisms-10-01695-f004:**
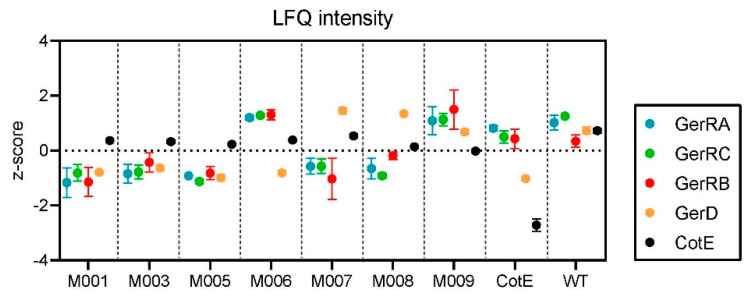
The abundances of three proteins encoded by the *gerR* operon (GerRA, GerRC and GerRB), GerD and CotE in spores of various strains. The z-scores of the LFQ intensities are shown to compare relative expression levels between different strains of a given protein; note that the quantitative data for GerRB are based on only a single peptide.

**Figure 5 microorganisms-10-01695-f005:**
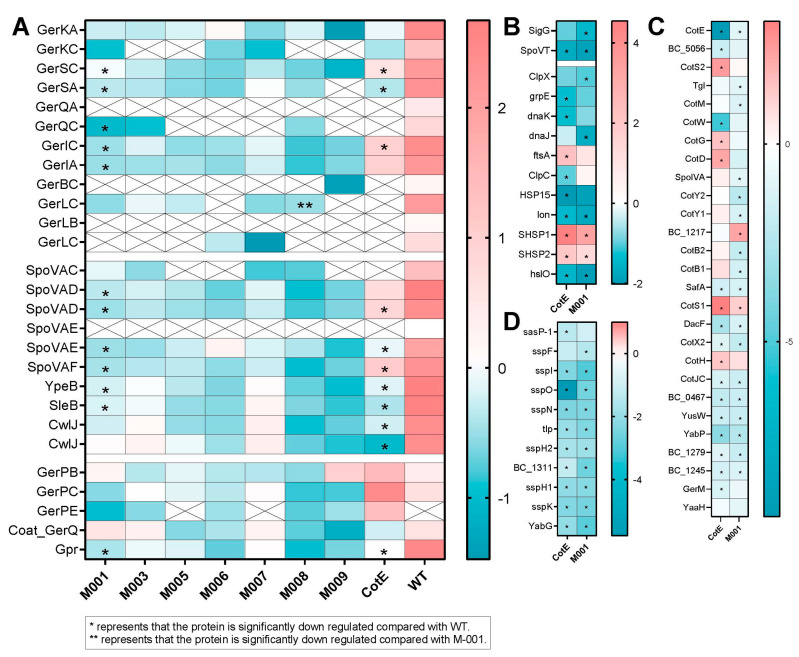
The heatmap of proteins associated with spore germination in spores of various strains (**A**). Color represents z-score of LFQ values; a cross means not detected. The log2 (fold changes) of SASPs (**D**), coat proteins (**C**) and heat shock proteins (**B**) are shown in a colored heatmap. Red represents elevated protein abundance and blue represents decreased LFQ intensities compared to WT spores.

**Figure 6 microorganisms-10-01695-f006:**
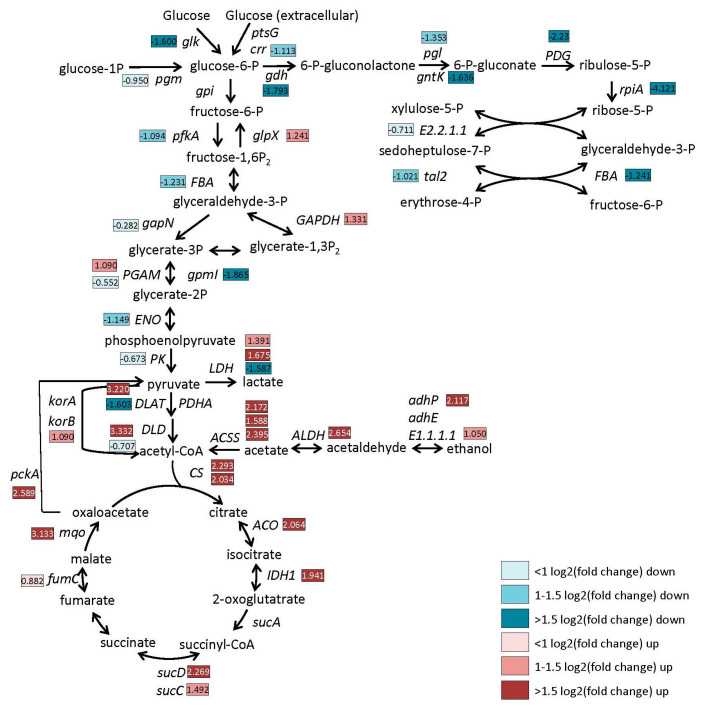
Enzyme levels (log2 (fold changes)) in central metabolic pathways of *B. cereus* strain M001 spores compared to levels in WT spores. Enzyme levels in blue are reduced, while those in red are increased. The multiple boxes next to some enzymes correspond to isoforms of the same enzyme. The regulation of the relevant genes and the enzymes for which they are coding are presented in Appendix A.

## Data Availability

Mass spectrometry data have been deposited and can be found at ProteomeXchange (PXD036145), and the Massive Repository for Mass Spectrometry data (MSV000090153).

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
