# Peer review of "Changes in the Spore Proteome of Bacillus cereus in Response to Introduction of Plasmids"

_microorganisms, 2022, doi:10.3390/microorganisms10091695_

Round 1
Reviewer 1 Report
This manuscript by Gao et al. elucidated that the presence of a plasmid required for molecular cloning in a host bacterium (here Bacillus cereus) causes broad changes in the organism’s proteome. Though the authors thus state the obvious, namely using appropriate controls for scientific experiments, the author were able to support this empirical truth with hard data.
General comments:
1) The manuscript is generally well written and mostly clear; there are only a variety of minor issues (see following comments below) that require a little further attention.
2) Please define all abbreviations in use at the first time of their occurrence in the text, so the non-expert reader can more easily understand the intended meaning.
3) Please italicize genes (and operons) throughout the text. Also make the distinction between gene (italicized with capital last letter) and protein (capital first and last letter, no italics). See also comment 11.
4) Please use the correct tense (past not present) for reporting results. There are many instances of this oversight in the Results section.
Specific comments:
5) Line 37: “…the spore core…”
6) L 51: “low mol wt nutrients”? (see comment (2) above).
7) L 66-71: I do not think it is necessary to explain GFP and its uses. Delete.
8) L 83: I suggest replacing “find” with “elucidate”.
9) L 91: “compared…plasmid-free…”
10) L 92: “…we were able to make…”
11) L 93: “protein expression”. Genes but not proteins are expressed. I suggest wording this as “heterologous protein production…and biosynthesis of specific GR…”
12) L 99: please make a reference to your Table 1 here. Also, please do this in L 129 and delete the first sentence in L 137.
13) L 144: it is a tad odd that the authors unnecessarily describe GFP in L 66-71 but then use a derivative of this protein (superfolder GFP) and leave the acronym sGFP unexplained (no capital “S”). BTW, the acronym sfGFP is more than twice more common (on google search) than sGFP.
14) L 158: close this line with a reference to Fig. 1B.
15) Figure 1: do the authors have an explanation for the delta in “numbers of proteins” identified” (y-axis) between wildtype vs all the eight mutants analyzed?
16) Figure 2: a tad larger size would be helpful for reading this figure (especially since there is unused space on the left and right margins. The same applies to Figs. 3 to 6.
17) L 182-197: this paragraph is the most “infamous” example for the issue described in comment (4).
18) L 183-186: the structure of this sentence is broken (no verb there for “combinations”). Maybe it would be best anyhow to separate this long sentence into two shorter ones.
19) L 216: here and elsewhere, I suggest using “compared to”.
20) L 219: replace “revival” with “germination”.
21) L 227: has the deletion (vulgo “knock out”) of the cotE gene in this mutant strain been confirmed prior to these experiments (e.g., by using the primers mentioned in reference 21)?
22) L 228: “…proteins were compared to…spores of strains…reduction in the recombinant strains…”
23) L240 and 275: delete the first “The”.
24) L 277-279: “Red represents…decreased LFQ…to WT spores.”
25) Fig.6: the black text in brown boxes is very hard to read. Can this be revised (e.g., using white text with brown fillings)?
26) L 313: it is odd that of all the acronyms used in Fig. only PEP is explained in the figure legend. Because I do not think it is necessary here to explain enzyme acronyms (such as GAPDH), I suggest you write out PEP directly in the figure. See also comment (3).
27) L 361-362: this sentence is unclear. Please rephrase.
28) L 381-382: delete one instance of this duplicate sentence.
29) L 386: fill in “…….” with actual data.
Reviewer 2 Report
This is a very interesting and well-written paper. Introduction is adequate and results and discussion sound good.
The authors investigated the effect on the spore proteome of introducing an empty plasmid or plasmids expressing fusion proteins and found that introduction of plasmid alone dramatically alters the spore-proteome. This result may be of great interest to the reader since researchers should keep in mind that molecular cloning can alter more than their intended targets and therefore they should performe adequate control experiments.
In my opinion two major point should be addrressed:
1. Among the proteins that are present and/or differentially expressed in the spore proteome the authors found proteins (HBl, motility etc...) whose effective function for the spore is questionable. The authors discuss this point stating that probabily the spore-proteome reflects the state that the vegetative cell had while undergoing sporulation. It is not clear (from the method section) if the authors performed any treatment (such as pasteurization) in order to remove any vegetative cell that can contaminate spore preparation. Indeed the rate of sporification for B. cereus is far from being 100% and presence of alive vegetative cells at the end of the sporification process cannot be excluded. In this case, some of the proteins that the authors include in the spore content may came from residual vegetative cells. This point should be considered both in the method section where more detailed procedures of sample preparation are required and in the result/discussion section.
2. Did the authors performed any observation regarding spore ability to germinate? Or did they test other properties? If cell trasformation may produce mutant aving some disturbation in spore proteome but there is no phisiological defect, this data can not be so important (this should be mentioned)
Minor revision:
lane 57: "to kill spore easily". Maybe is better "to kill germinated spores easily" or "to kill vegetative cells easily".
lane 105: washing or washed?
lane 101 102: please specify when antibiotics were used.
lane 105: did the authors performed any treatment to remove residual intact cells?
Author Response
"Please see attachment".
